# An Epidemiological Systematic Review with Meta-Analysis on Biomarker Role of Circulating MicroRNAs in Breast Cancer Incidence

**DOI:** 10.3390/ijms24043910

**Published:** 2023-02-15

**Authors:** Lisa Padroni, Laura De Marco, Lucia Dansero, Valentina Fiano, Lorenzo Milani, Paolo Vasapolli, Luca Manfredi, Saverio Caini, Claudia Agnoli, Fulvio Ricceri, Carlotta Sacerdote

**Affiliations:** 1Unit of Cancer Epidemiology, Città della Salute e della Scienza University-Hospital and Center for Cancer Prevention (CPO), Via Santena 7, 10126 Turin, Italy; 2Centre for Biostatistics, Epidemiology and Public Health (C-BEPH), Department of Clinical and Biological Sciences, University of Turin, 10043 Orbassano, Italy; 3Unit of Cancer Epidemiology, Department of Medical Sciences, University of Turin, 10126 Turin, Italy; 4Institute for Cancer Research, Prevention and Clinical Network (ISPRO), 50139 Florence, Italy; 5Epidemiology and Prevention Unit, Fondazione IRCCS Istituto Nazionale dei Tumori, 20133 Milan, Italy; 6Unit of Epidemiology, Regional Health Service ASL TO3, 10095 Grugliasco, Italy

**Keywords:** breast cancer, microRNA, miRNA, serum, plasma, blood

## Abstract

Breast cancer (BC) is a multifactorial disease caused by an interaction between genetic predisposition and environmental exposures. MicroRNAs are a group of small non-coding RNA molecules, which seem to have a role either as tumor suppressor genes or oncogenes and seem to be related to cancer risk factors. We conducted a systematic review and meta-analysis to identify circulating microRNAs related to BC diagnosis, paying special attention to methodological problems in this research field. A meta-analysis was performed for microRNAs analyzed in at least three independent studies where sufficient data to make analysis were presented. Seventy-five studies were included in the systematic review. A meta-analysis was performed for microRNAs analyzed in at least three independent studies where sufficient data to make analysis were presented. Seven studies were included in the MIR21 and MIR155 meta-analysis, while four studies were included in the MIR10b metanalysis. The pooled sensitivity and specificity of MIR21 for BC diagnosis were 0.86 (95%CI 0.76–0.93) and 0.84 (95%CI 0.71–0.92), 0.83 (95%CI 0.72–0.91) and 0.90 (95%CI 0.69–0.97) for MIR155, and 0.56 (95%CI 0.32–0.71) and 0.95 (95%CI 0.88–0.98) for MIR10b, respectively. Several other microRNAs were found to be dysregulated, distinguishing BC patients from healthy controls. However, there was little consistency between included studies, making it difficult to identify specific microRNAs useful for diagnosis.

## 1. Introduction

Breast cancer (BC) is the most frequently diagnosed cancer in Europe, accounting for 13% of all new cancer cases [1].

BC is a multifactorial disease caused by the interaction between genetic predisposition and environmental exposures [2]. The environmental exposures include several modifiable risk factors such as overweight or obesity (post-menopausal), use of menopausal hormone therapy, a low level of physical activity, consumption of alcohol, cigarette smoking, shift work, and some reproductive factors [2]. A genetic predisposition or family history account for about 10%, with some geographical variations [2]. The most common are germline mutations, such as BRCA1, BRCA2, PALB2, ATM, and TP53 genes, among others [3,4].

MicroRNA are a group of short noncoding regulatory RNAs that modulate gene expression at the post transcriptional level [5]. The dysregulation of microRNAs is linked to many human diseases, including cancer. Cell-free circulating microRNAs probably released from cells in lipid vescicles, microvescicles, or exosomes have been detected in peripheral blood circulation [6].

Due to the stability and resistance to the endogenous RNase activity, microRNAs have been investigated as diagnostic biomarkers of BC. Accessing circulating BC biomarkers from peripheral blood (through the so-called liquid biopsy) is a promising non-invasive and cost-effective procedure [7]. In fact, dysregulated microRNAs have both oncogenic and tumour-suppressing actions, depending on their targets [7]. This is a complex matter because some dysregulations of microRNAs seem to be common in most cancers, possibly due to their role in cancer-associated biological processes and not in aetiology targets [7].

Circulating microRNA may reflect the response of the organism to environmental exposures, as well as early signs of disease.

This review aims to report the potential use of altered circulating microRNA levels in the diagnosing of BC, paying special attention to methodological problems in this research field.

## 2. Materials and Methods

The protocol of this review was registered in the international database of prospective registered systematic reviews (PROSPERO 2022; CRD42022354439). The workflow and methodology were based on the guidelines of Preferred Reporting Items for Systematic Reviews and Meta-Analyses of Diagnostic Test Accuracy (PRISMA-DTA) [8].

### 2.1. Publication Search

We conducted a comprehensive literature search in PubMed, Cochrane Library, EMBASE, Google Scholar, and NCBI PubMed Central until 31 August 2022 to identify relevant studies. The article search was performed using the following search strategy:

((Circulating) AND (microRNA OR miRNA) AND (breast AND Cancer)) NOT (cells) NOT (tissue) AND ((English[Filter]) AND (Humans[Filter]) AND (“31 August 2022”[Date—Publication]))

Furthermore, other relevant studies were identified by manually searching for references of eligible publications.

### 2.2. Inclusion and Exclusion Criteria

Studies were considered eligible for the systematic review if they met the following criteria: (1) The study includes patients with BC and healthy controls; (2) The levels of one or more microRNAs were measured in blood, serum, or plasma; (3) They presented sufficient data to collect the number of patients and a measure of diagnostic performance (e.g., sensibility and sensitivity, or fold change) or a measure of association (e.g., Odds Ratio or Relative Risk).

Studies were included in the meta-analysis if there were at least three studies focused on the same microRNA, they met criteria (1), (2), and (3), and the frequencies of true positives (TP), false positives (FP), true negatives (TN), and false negatives (FN) could be directly or indirectly extracted.

Studies were excluded if they were reviews, meta-analysis, letters, commentaries, or abstracts presented in conferences; lacking sufficient data; duplication of previous publications; or languages other than English.

### 2.3. Data Extraction

After the selection of studies was made, other relevant studies were searched from the references in the articles. According to inclusion criteria, data were extracted by two independent authors (LP and CS). Disagreements were solved through face-to-face discussion and consensus. Extracted data form included: first author’s name and reference, country, sample size, biological sample (plasma, serum, or blood), microRNA, cut-off value, AUC value (95% CI), sensitivity (95% CI), specificity (95% CI), fold change (95% CI), *p*-value, microRNA source (candidate or discovery if found in a screening phase), and expression (upregulation or downregulation). Diagnostic performance data were extracted or calculated for the studies included in the meta-analysis (FP, FN, TP, TN).

### 2.4. Quality Assessment

We estimated the quality of each study using the revised Quality Assessment of Diagnostic Accuracy Studies (QUADAS-2) by two independent authors (LP and CS) [9].

### 2.5. Statistical Analysis

The STATA17.0 software was used to realize the statistical analyses.

Descriptive statistics on directions of microRNA expression are displayed in a pyramidal graph by type of specimen. Direction of microRNA expression was defined as the direction of the ratio between the microRNA concentration in breast cancer cases and the microRNA concentration in breast cancer controls.

For each study included in the meta-analysis, we built a contingency table to be used to carry out the meta-analysis. After selecting suitable studies, forest plot, and summary receiver operating characteristic curve (SROC), with the pooled sensitivity and specificity, were built for each microRNA [10,11]. We analyzed the heterogeneity between studies using the I2 statistics. Funnel plots were used to evaluate publication bias [12].

The analyses were repeated on subgroups as sensitivity analysis. Subgroups analyses were based on specimen type, ethnicity, and quality of the study (by QUADAS-2 score).

## 3. Results

In total, 149 eligible studies were obtained from online database searching after automation screening. After a manual check of titles and abstracts, 47 papers were excluded because of the type of study (review, prognostic studies) or they were out of topic. After screening full texts, 24 publications were excluded because they did not satisfy the inclusion criteria. Finally, 75 publications were considered in this review [13,14,15,16,17,18,19,20,21,22,23,24,25,26,27,28,29,30,31,32,33,34,35,36,37,38,39,40,41,42,43,44,45,46,47,48,49,50,51,52,53,54,55,56,57,58,59,60,61,62,63,64,65,66,67,68,69,70,71,72,73,74,75,76,77,78,79,80,81,82,83,84,85,86,87]. The flow-chart of the excluded papers is presented in Figure 1. The main characteristics of the studies were summarized in Table 1.

The publications involved a total of 6380 BC cases and 4517 health controls. Studies with two different approaches were included in this review: (I) The validation phase of studies where investigated microRNA were selected on the basis of a previous discovery phase (N = 26); (II) Candidate studies where microRNA were selected based on a priori knowledge (N = 50).

The studies with a number of BC cases ≥ 100 were only 20/75 (≈26%). The studies were conducted in China (N = 19), Germany (N = 9), Egypt (N = 6), USA (N = 6), Iran (N = 4), Ireland (N = 4), Spain (N = 3), Belgium (N = 2), Czech Republic (N = 2), India (N = 2), Indonesia (N = 2), Mexico (N = 2), Singapore (N = 2), and others (N = 12). Notably, the majority of studies were conducted in the Eastern Asian population (N = 29), while the majority of the others were conducted among European or white U.S. populations. Black and Hispanic populations are very little represented in microRNA and breast cancer studies.

Some studies were performed in serum (N = 44), while others were performed in plasma (n = 21), whole blood (N = 7), plasma and serum (N = 2), or blood, serum, and plasma (N = 1).

Among the 75 studies included in the review, 53 studies conducted multiple microRNA assays, while the other 22 studies focused on single microRNA assay (covering in total 141 microRNAs).

In the Appendix A, the clinical information on cases is presented for each study.

The results of the studies included in the review are presented in Table 2. The microRNAs that were proposed as a biomarker of BC in at least two independent clinical studies in the same biological specimen (serum, plasma, or whole blood) were: MIR17, MIR21, MIR24, MIR145, MIR155, MIR195, MIR202, MIR222, MIR335, MIR373, MIR425, MIR652, MIR10b, MIR29b, MIR34a, MIR92a, MIR148b, MIR181a, MIR199, and MIR1246. The direction of the ratio of microRNA concentrations in breast cancer cases versus controls was generally coherent for MIR21 (12 up versus 1 down), MIR155 (14 up versus 1 down), MIR10b (5 up, only serum), MIR373 (3 up, only serum), MIR652 (3 down, only serum), MIR425 (2 down, only serum), MIR29a (2 up, only serum), and MIR148b (2 down, only serum). The direction was not coherent for MIR145 (5 up versus 4 down), MIR17, MIR24, MIR195, MIR202, MIR222, MIR335, MIR451, MIR1246, MIR34a, MIR92a, MIR181a, and MIR199a (Figure 2).

Not all the studies presented data on AUC and/or sensitivity and specificity or fold change (N = 26 articles did not report sensitivity/specificity or AUC measures; N = 48 not reported fold change measure for single miRNAs).

A meta-analysis was performed only for microRNAs analysed in at least three independent studies where sufficient data to make an analysis were presented.

The results of the meta-analysis on MIR21 (upregulated), showed an overall sensitivity of 0.86 (95%CI 0.76–0.93) and a specificity of 0.84 (95%CI 0.71–0.92) (Figure 3).

The pooled sensitivity and specificity of MIR155 (upregulated) were respectively 0.83 (95%CI 0.72–0.91) and 0.90 (95%CI 0.69–0.97) (Figure 4).

The meta-analysis results on the accuracy of MIR10b demonstrated a very low sensitivity (0.56 95%CI 0.32–0.71) and a high specificity (0.95 95%CI 0.88–0.98) (Figure 5).

Meta-analysis results for MIR34a and MIR195 were presented in the Appendix A.

The shape of the funnel plot showed asymmetry in the analyses of MIR21, MIR155, and the overall microRNAs included in the meta-analysis, implying the presence of a publication bias in the analysis of the remaining circulating microRNAs (Figure 6).

In general, a quality assessment with QUADAS detected a low quality of the studies because of an inadequate sample size and a low attention to the study design, the choice of controls, and the possible confounders (Figure 7 and Appendix A).

## 4. Discussion

In the present study, we systematically reviewed clinical studies on microRNA for the diagnosis of BC, exploring possible links of most associated microRNA with hallmarks of BC.

Increasing evidence has demonstrated that microRNA may function as either a tumor suppressor or a promoter in a variety of cancers. The association of obesity and inflammation with microRNA has also been proposed [88,89]. The identification of microRNAs that could simultaneously be associated to BC and other hallmarks of cancer could be considered a meet-in-the-middle biomarker, which is useful to disentangle the role of involved factors and to hypothesize a biological pathway from exposures to disease [90].

Only few microRNAs were analyzed in more than two independent studies that presented in the results the essential data to be included in this meta-analysis. Among them, the most interesting microRNAs in terms of coherence among studies in the regulation direction and of results of diagnostic accuracy were two upregulated microRNAs: MIR21 and MIR155.

The MIR21 is an onco-microRNA that inhibits several tumor-suppressor genes (such as PTEN) and promotes cell growth invasion, apoptosis, and immune dysregulation [91]. A significant interaction between obesity and the expression of MIR21 and MIR155 consisting of obesity reducing the expression of these microRNAs in control women was described [89]. MIR155 is another oncogenic microRNA that regulates several signaling pathways related to cell growth [92], and it is also known to target BRCA1 [93]. It has also an important role in reducing inflammation, observed both in vitro and in vivo [94].

We observed little consistency with respect to the circulating microRNAs identified by different studies. This could be due also because of the different method used in the choice of microRNAs, lack of standardization of techniques (different sample retrieval and conservation, laboratory techniques, microRNAs measurements and normalization, cut-off), inconsistent selection of patients, low abundance, small samples size, and inadequate statistical analysis.

The majority of studies analyzed microRNA concentration in serum, but others used plasma or whole blood. Most of the microRNAs in the serum showed higher concentrations than the corresponding plasma samples [95], and some of the discrepancies in the direction of microRNAs presented in Table 1 could be due to the different types of samples.

Furthermore, the most frequent method used to quantify circulating candidate microRNAs, or to validate microRNAs, is the quantitative reverse transcription (PCR RT-qPCR); only two studies have introduced the use of the digital droplet [46,76]. All these techniques have a high sensitivity in detecting a large number of microRNAs at the same time. However, a poor agreement among different microRNA measurement platforms has been reported [96].

For microRNA measurement, data normalization is still a challenge, and this could be another source of variability in the results [95]. Furthermore, the cut-off values of considered microRNAs to calculate sensibility and specificity based on different ROC curves were not uniform, which may contribute to the observed heterogeneity.

The sample sizes in most studies are relatively small, and very few studies included an adequate group of controls that were collected from the same population, rather than cases, and matched at least for age.

In the reviewed articles, two categories of studies on microRNA and BC diagnosis were present: (i) Studies with an agnostic approach based on microRNA profiling (using different array platforms), usually followed by a validation phase in a different population with a more sensitive technique of most promising selected microRNA; (ii) Studies with a Bayesian approach based on microRNAs candidates, selected on the basis of previous biological knowledge, positive results in other studies, or in studies on other cancers.

In the present review, both the categories of studies were included, but only the validation phase of the microRNA profiling studies was described.

Finally, due to genetic heterogeneity, a difference in identified microRNA may be present among different ethnicities.

The result is a high number of microRNA identified as possibly related to BC status, but very few of them were replicated in other studies and other populations.

This review is a very comprehensive collection of studies on circulating microRNA and breast cancer. However, it presents several limitations. In fact, in all the studies, the origin of microRNAs has not been verified, and the contribution of breast cancer tissue has not been verified. There was a high heterogeneity among studies, probably due to different ethnic populations, small sample sizes, different types of sample and laboratory techniques, different statistical analysis, and different cut-offs.

## 5. Conclusions

The effort of this review has been devoted to exploring the most important microRNA involved in BC pathogenesis.

We found a list of microRNAs possibly involved in the breast cancer pathogenic pathway. Anyway, an effort must be done to try to standardize the microRNA research, with more robust study design, analytical strategies, and a better reporting of results in the published articles. New studies nested in population cohorts are needed to analyze microRNA in pre-diagnostic blood samples in order to strengthen the evidence of the association with breast cancer.

## Figures and Tables

**Figure 1 ijms-24-03910-f001:**
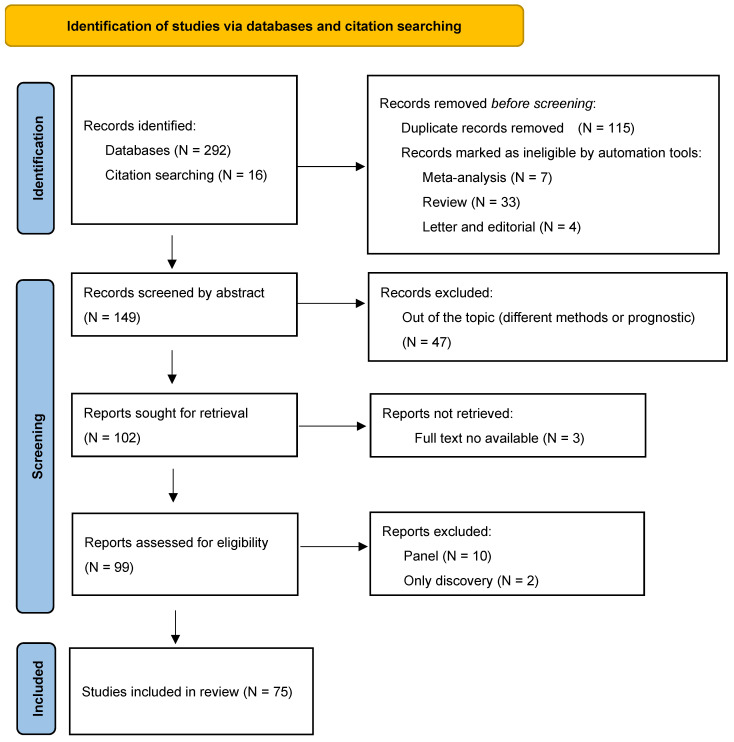
The flow chart of identification, screening, and eligibility of the included studies [8].

**Figure 2 ijms-24-03910-f002:**
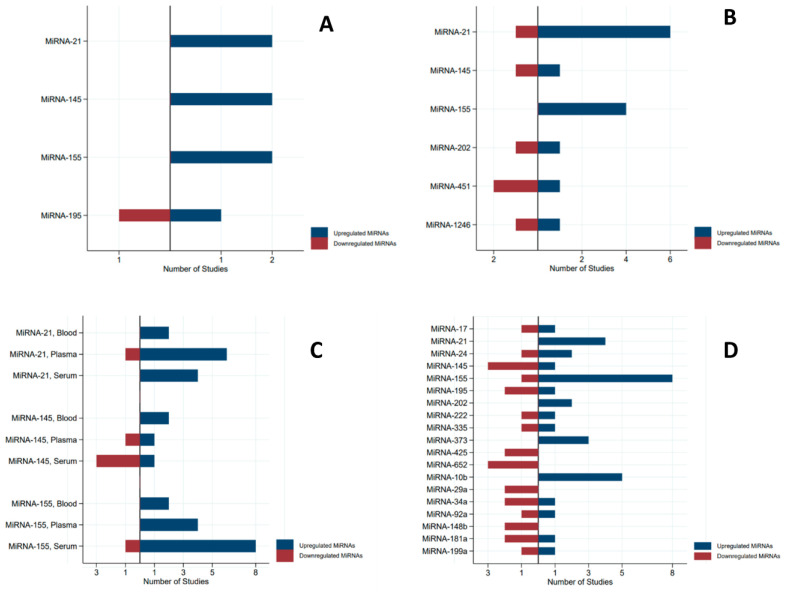
Pyramidal graph of the direction of miRNA expression (microRNA concentration in breast cancer cases versus controls) by type of specimens (only microRNAs that were analysed in two or more independent studies). (**A**) = whole blood, (**B**) = plasma; (**C**) = all specimens; (**D**) = serum.

**Figure 3 ijms-24-03910-f003:**
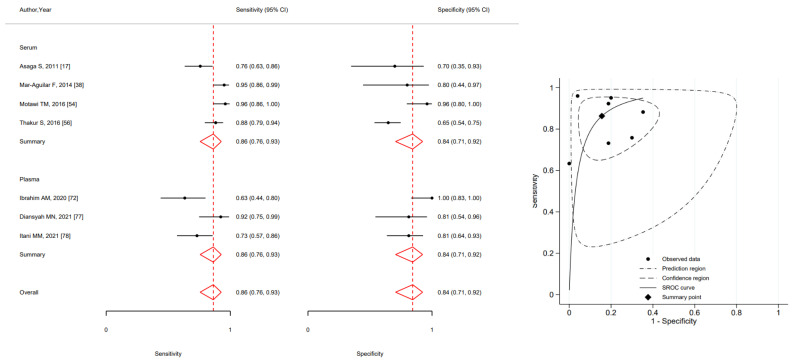
Forest plot of included studies assessing the sensitivity and specificity by type of specimen and summary receiver operating characteristic curve (SROC) of MIR21 in breast cancer diagnosis (squares shows sensitivity and specificity, respectively; red diamonds show pooled effect; error bars represents 95% confidence interval) [17,38,54,56,72,77,78].

**Figure 4 ijms-24-03910-f004:**
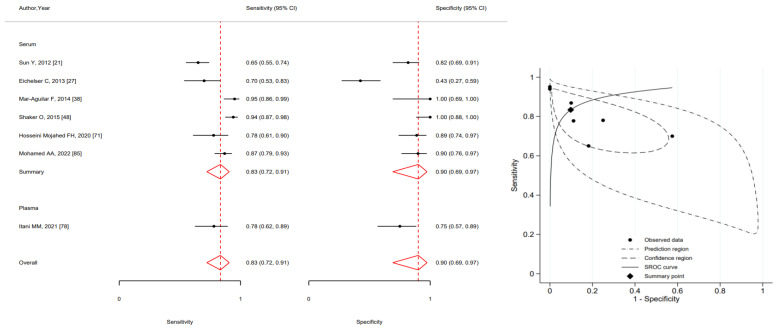
Forest plot of included studies assessing the sensitivity and specificity by type of specimen and summary receiver operating characteristic curve (SROC) of MIR155 in breast cancer diagnosis (squares shows sensitivity and specificity, respectively; red diamonds show pooled effect; error bars represents 95% confidence interval) [21,27,38,48,71,78,85].

**Figure 5 ijms-24-03910-f005:**
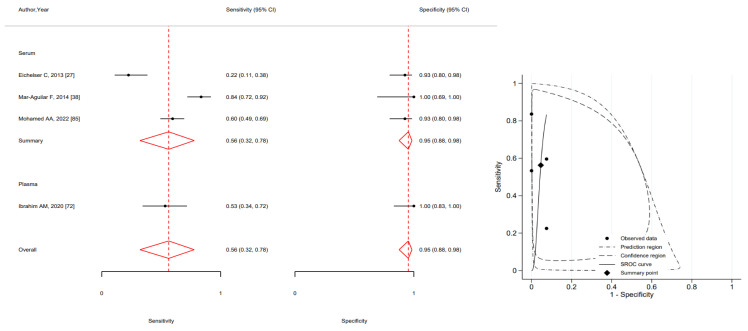
Forest plot of included studies assessing the sensitivity and specificity by type of specimen and summary receiver operating characteristic curve (SROC) of MIR10b in breast cancer diagnosis (squares shows sensitivity and specificity, respectively; red diamonds show pooled effect; error bars represents 95% confidence interval) [27,38,72,85].

**Figure 6 ijms-24-03910-f006:**
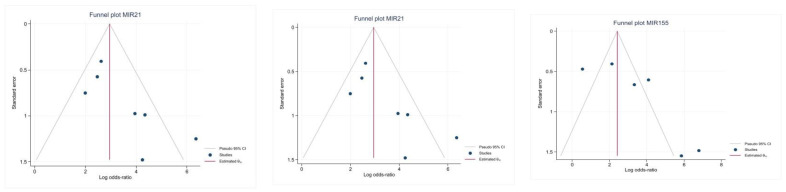
Evaluation of publication bias of all reported microRNAs presented as funnel plots.

**Figure 7 ijms-24-03910-f007:**
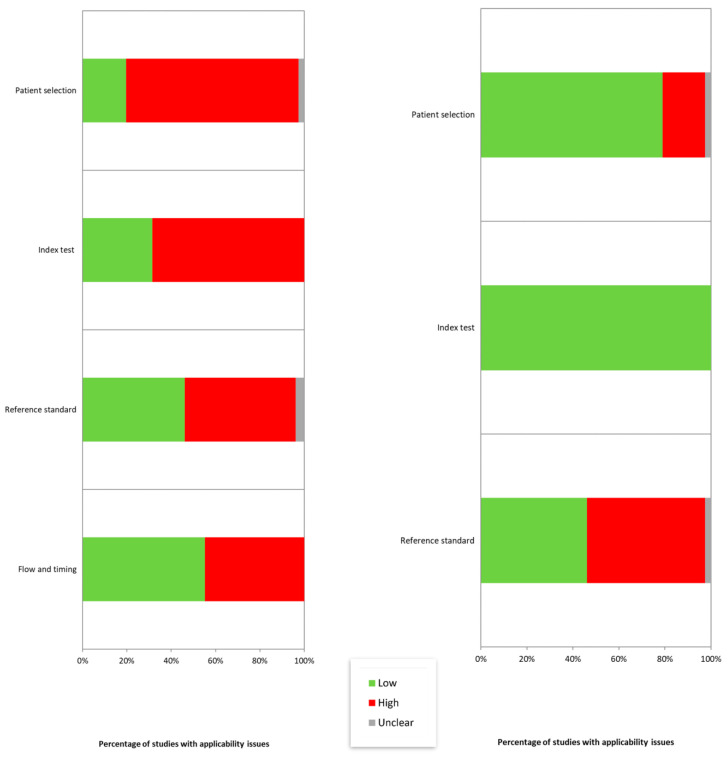
Quality assessment with the QUADAS-2 tool [9].

**Table 1 ijms-24-03910-t001:** General characteristics of the studies included in systematic review on the role of microRNA in breast cancer diagnosis.

First Author, Year	Country	Type	Specimen Source	Cases Size	Controls Size	MIR	Internal Reference
Zhu W, 2009 [13]	USA	Candidate	Serum	13	8	16	MIR18
						145	
						155	
Heneghan H, 2010 [14]	Ireland	Candidate	Blood	83	44	21	MIR16
						145	
						155	
						195	
						10b	
						Let-7a	
Roth C, 2010 [15]	Germany	Candidate	Serum	59	29	141	MIR16
						155	
						10b	
						34a	
Wang F, 2010 [16]	China	Candidate	Serum	68	40	21	MIR16
						126	
						155	
						335	
						106a	
						199a	
Asaga S, 2011 [17]	USA	Validation	Serum	62	10	21	MIR16
Guo LJ, 2012 [18]	China	Candidate	Serum	152	75	181a	MIR16
Schrauder MG, 2012 [19]	Germany	Validation	Serum	24	24	202	MIR16
						718	
Schwarzenbach H, 2012 [20]	Germany	Candidate	Serum	34	53	21	MIR16
						214	
						19a	
						20a	
Sun Y, 2012 [21]	China	Candidate	Serum	103	55	155	MIR39
van Schooneveld E, 2012 [22]	Belgium	Candidate	Serum	75	20	215	
						299	
						411	
						452	
Wu Q, 2012 [23]	China	Validation	Serum	50	50	222	
Zhao FL, 2012 [24]	China	Candidate	Serum	122	59	10b	MIR16
Chan M, 2013 [25]	Singapore	Validation	Serum	132	101	1	MIR103, MIR191
						133a	
						133b	
						92a	
Cuk K, 2013 [26]	Germany	Validation	Plasma	127	80	409	MIR39
						801	
						148b	
						376c	
Eichelser C, 2013 [27]	Germany	Candidate	Serum	40	40	17	MIR16
						93	
						155	
						373	
						10b	
						34a	
Godfrey AC, 2013 [28]	USA	Validation	Serum	5	5	222	MIR1825
						181a	
						18a	
Kumar S, 2013 [29]	India	Candidate	Plasma	14	8	21	MIR16
						146a	
Ng EKO, 2013 [30]	China	Validation	Plasma	170	100	16	RNU6B
						21	
						145	
						451	
Si H, 2013 [31]	China	Validation	Serum	100	20	21	MIR16
						92a	
Wang PY, 2013 [32]	China	Candidate	Serum	46	58	182	5S rRNA
Zeng RC, 2013 [33]	China	Candidate	Plasma	100	64	30a	MIR16
Eichelser C, 2014 [34]	Germany	Candidate	Serum	168	28	101	MIR16, MIR 484
						372	
						373	
Hamdi K, 2014 [35]	Tunisia	Candidate	Serum	20	20	24	RNU48
						320	
						335	
						337	
						451	
						486	
						548	
						15a	
						29a	
						30b	
						342-3p	
						342-5p	
Joosse SA, 2014 [36]	Germany	Candidate	Serum	102	37	202	MIR16
						Let-7b	
Kodahl AR, 2014 [37]	Denmark	Validation	Serum	60	51	107	
						139	
						143	
						145	
						365	
						425	
						133a	
						15a	
						18a	
Mar-Aguilar F, 2014 [38]	Mexico	Validation	Serum	61	10	21	MIR18S
						145	
						155	
						191	
						382	
						10b	
						125b	
McDermott AM, 2014 [39]	Ireland	Validation	Serum	44	46	223	MIR16
						652	
						181a	
						29a	
Shen J, 2014 [40]	USA	Validation	Plasma	50	50	409	MIR93
						133a	
						148b	
Sochor M, 2014 [41]	Chez Republic	Candidate	Serum	63	21	24	Let-7a
						155	
						181b	
						19a	
Zearo S, 2014 [42]	Australia	Validation	Serum	98	25	484	
Zhao FL, 2014 [43]	China	Candidate	Serum	210	102	195	MIR16
Antolin S, 2015 [44]	Spain	Candidate	Blood, serum and plasma	57	20	141	5S, U6 sn
						200c	
Li XX, 2015 [45]	China	Candidate	Serum	90	64	Let-7c	5SrRNA
Mangolini A (A), 2015 [46]	Italy	Candidate	Serum	28	27	145	MIR39
						425	
						652	
						10b	
						148b	
Mangolini A (B), 2015 [46]	USA	Candidate	Serum	59	35	145	MIR39
						425	
						652	
						10b	
						148b	
Matamala N, 2015 [47]	Spain	Validation	Plasma	114	116	21	MIR103a
						96	
						505	
						125b	
Shaker O, 2015 [48]	Egypt	Candidate	Serum	100	30	155	SNORD
						197	
						205	
						29b	
Zhang L, 2015 [49]	China	Validation	Serum	76	52	424	MIR132
						199a	
						29c	
Frères P, 2016 [50]	Belgium	Validation	Plasma	108	88	16	Median of 50 mirna
						22	
						103	
						107	
						148a	
						19b	
						Let-7d	
						Let-7i	
Fu L, 2016 [51]	China	Candidate	Serum	100	40	184	
						382	
						598	
						1246	
Hamam R, 2016 [52]	Saudi Arabia	Validation	Serum and Plasma	46	50	188	MIR21
						1202	
						1207	
						1225	
						1290	
						3141	
						4270	
						4281	
						642b	
Hannafon BN, 2016 [53]	USA	Candidate	Plasma	16	42	21	MIR54
						122	
						1246	
						Let-7a	
Motawi TM, 2016 [54]	Egypt	Candidate	Serum	50	25	21	REF SNORD 62
						221	
Shimomura A, 2016 [55]	Japan	Validation	Serum	1206	1343	1246	MIR149
						1307	
						4634	
						6861	
						6875	
Thakur S, 2016 [56]	India	Candidate	Serum	85	85	21	Sn U6
						145	
						195	
						210	
						221	
						Let-7a	
Gao S, 2017 [57]	USA	Validation	Plasma	75	50	155	RNU6B
Zhang K, 2017 [58]	China	Validation	Blood	15	13	96	MIR16
						182	
						942	
						30b	
						374b	
Heydari N, 2018 [59]	Iran	Candidate	Serum	40	40	140	MIR16
Zaleski M, 2018 [60]	Germany	Validation	Plasma	55	28	21	MIR16
						92	
						155	
						222	
						34a	
						Let-7c	
Kaharam M, 2019 [61]	Germany	Validation	Blood	21	21	101-3p	RNU48
						126-3p	
						126-5p	
						144-3p	
						144-5p	
						301a	
						664b	
McAnena P, 2019 [62]	Ireland	Validation	Blood	31	34	195	MIR16, MIR425
						331	
						181a	
Peña-Cano MI, 2019 [63]	Mexico	Candidate	Serum	50	50	17	MIR26b
						195	
						221	
Raheem AR, 2019 [64]	Iraq	Candidate	Serum	30	30	34a	MIRU6
Soleimanpour E, 2019 [65]	Iran	Candidate	Plasma	30	25	21	MIR5s
						155	
Anwar SL, 2020 [66]	Indonesia	Candidate	Plasma	102	15	155	Sp6
Arabkari V, 2020 [67]	Ireland	Validation	Blood	38	20	16	MIR1, MIR16
						21	
						145	
						155	
						195	
						486	
						181a	
						451a	
Ashirbekov Y, 2020 [68]	Kazakhstan	Candidate	Plasma	35	33	16	MIR222
						21	
						29	
						145	
						191	
						210	
						222	
Guo H, 2020 [69]	China	Validation	Plasma	39	40	21	cel-39
						1273g	
Holubekova V, 2020 [70]	Slovakia	Validation	Plasma	65	34	484	MIR16, MIR103a
						1260a	
						130a	
						99a	
Hosseini Mojahed FH, 2020 [71]	Iran	Candidate	Serum	36	36	155	
Ibrahim AM, 2020 [72]	Egypt	Candidate	Plasma	30	20	21	MIR16
						145	
						10b	
						181a	
						Let-7	
Jang JY, 2020 [73]	Korea	Validation	Plasma	80	56	21	
						24	
						202	
						206	
						223	
						373	
						1246	
						6875	
						219b	
Kim J, 2020 [74]	South Korea	Candidate	Plasma	30	30	202	
Pastor-Navarro B, 2020 [75]	Spain	Candidate	Serum	45	16	21	MIR16, MIR1228
						155	
						205	
Bakr NM, 2021 [76]	Egypt	Validation	Blood	196	49	373	
Diansyah MN, 2021 [77]	Indonesia	Candidate	Plasma	26	16	21	MIR16
Itani MM, 2021 [78]	Lebanon	Candidate	Plasma	41	32	21	
						139	
						145	
						155	
						425	
						451	
						130a	
						23a	
Mohmmed EA, 2021 [79]	Egypt	Candidate	Serum and Plasma	50	30	106a	
Nashtahosseini Z, 2021 [80]	Iran	Candidate	Serum	40	40	210	MIR16
						660	
Zhang K, 2021 [81]	China	Validation	Blood	68	13	185	
						362	
						106b	
						142-3p	
						142-5p	
						26b	
Zhao T, 2021 [82]	China	Candidate	Serum	88	40	25	MIR39
Li X, 2022 [83]	China	Candidate	Serum	49	49	9	MIR16
						17	
						148a	
Liu H, 2022 [84]	China	Candidate	Serum	112	59	103a	U6 sn
Mohamed AA, 2022 [85]	Egypt	Candidate	Serum	99	40	155	RNU6
						373	
						10b	
						34a	
Zavesky L, 2022 [86]	Czech Republic	Validation	Plasma	52	46	451a	MIR590, MIR19a, MIR222
						548b	
Zou R, 2022 [87]	Mix	Validation	Serum	177	197	24	MIR128, MIR652, MIR106b
						324	
						377	
						497	
						125b	
						133a	
						19b	
						374c	

**Table 2 ijms-24-03910-t002:** Summary of the results of the studies included in the systematic review on the role of microRNA in breast cancer diagnosis.

First Author, Year	Specimen Source	MiR	Direction	Cut_Off (ng/mL)	AUC	Sens	Spec	Fold Change
Zhu W, 2009 [13]	Serum	16	Up					
		145	Up					
		155	Down					
Heneghan H, 2010 [14]	Blood	21	Up					
		145	Up					
		155	Up					
		195	Up		0.94 (0.91–0.97)	87.70	91.00	25.00
		10b	Down					
		Let-7a	Up					
Roth C, 2010 [15]	Serum	141						
		155	Up					1.60
		10b						
		34a						
Wang F, 2010 [16]	Serum	21	Up					2.50
		126	Down					2.00
		155	Up					3.50
		335	Up					2.00
		106a	Up					1.90
		199a	Down					2.00
Asaga S, 2011 [17]	Serum	21	Up	3.30	0.72	75.00	67.00	
Guo LJ, 2012 [18]	Serum	181a	Down	0.74	0.67 (0.60–0.74)	70.70	59.90	0.36
Schrauder MG, 2012 [19]	Serum	202	Up		0.68			19.38
		718	Down		0.77			5.44
Schwarzenbach H, 2012 [20]	Serum	21			0.85 (0.78–0.91)			
		214			0.92 (0.88–0.97)			
		19a						
		20a			0.68 (0.59–0.77)			
Sun Y, 2012 [21]	Serum	155	Up	1.91	0.80 (0.65–0.82)	65.00	81.80	2.94
van Schooneveld E, 2012 [22]	Serum	215	Up					
		299	Down					
		411	Down					
		452	Down					
Wu Q, 2012 [23]	Serum	222	Up	0.01	0.67 (0.57–0.78)	74.00	60.00	
Zhao FL, 2012 [24]	Serum	10b	Up					
Chan M, 2013 [25]	Serum	1	Up					2.67
		133a	Up					2.62
		133b	Up					2.41
		92a	Up					1.32
Cuk K, 2013 [26]	Plasma	409	Up		0.66 (0.59–0.74)			
		801	Up		0.64 (0.56–0.72)			
		148b	Up		0.65 (0.58–0.73)			
		376c	Up		0.66 (0.59–0.74)			
Eichelser C, 2013 [27]	Serum	17	Down		0.68	18.80	100.00	
		93	Up		0.70	44.90	100.00	
		155	Up		0.78	70.60	42.70	
		373	Up		0.88	76.60	100.00	
		10b	Up			21.80	92.10	
		34a	Up		0.64	59.80	76.00	
Godfrey AC, 2013 [28]	Serum	222	Down					
		181a	Up					
		18a	Up					
Kumar S, 2013 [29]	Plasma	21	Up					
		146a	Up					
Ng EKO, 2013 [30]	Plasma	16	Up		0.91 (0.87–0.95)			
		21	Up		0.81 (0.74–0.88)			
		145	Down		0.63 (0.52–0.74)			
		451	Up		0.94 (0.91–1.00)			
Si H, 2013 [31]	Serum	21	Up		0.93 (0.89–0.92)			
		92a	Down		0.92 (0.87–0.97)			
Wang PY, 2013 [32]	Serum	182	Up					
Zeng RC, 2013 [33]	Plasma	30a	Down	0.01	0.76 (0.68–0.83)	74.00	65.60	
Eichelser C, 2014 [34]	Serum	101	Up					
		372	Up					
		373	Up					
Hamdi K, 2014 [35]	Serum	24	Down					
		320	Down					
		335	Down					
		337	Down					
		451	Down					15.80
		486	Down					
		548	Down					
		15a	Down					
		29a	Down					
		30b	Down					
		342-3p	Down					
		342-5p	Down					
Joosse SA, 2014 [36]	Serum	202	Up					
		Let-7b	Up					
Kodahl AR, 2014 [37]	Serum	107						0.66
		139						1.44
		143						1.65
		145						1.56
		365						1.88
		425						0.84
		133a						1.68
		15a						1.84
		18a						0.65
Mar-Aguilar F, 2014 [38]	Serum	21		6.48	0.95 (0.91–0.99)	94.40	80.00	
		145		15.93	0.98 (0.95–1.00)	94.40	100.00	
		155		7.92	0.99 (0.99–1.00)	94.40	100.00	
		191		4.85	0.79 (0.71–0.88)	72.20	90.00	
		382			0.97 (0.94–1.00)	94.40	90.00	
		10b		59.22	0.95 (0.91–0.99)	83.30	100.00	
		125b		8.46	0.95 (0.91–0.99)	88.90	80.00	
McDermott AM, 2014 [39]	Serum	223	Down					
		652	Down					
		181a	Down					
		29a	Down					
Shen J, 2014 [40]	Plasma	409						
		133a						8.30
		148b						5.10
Sochor M, 2014 [41]	Serum	24	Up					
		155	Up					
		181b	Up					
		19a	Up					
Zearo S, 2014 [42]	Serum	484						1.60
Zhao FL, 2014 [43]	Serum	195	Down	0.50	0.86 (0.82–0.90)	69.00	89.20	2.38
Antolin S, 2015 [44]	Blood, serum and plasma	141						
		200c	Down		0.79	90.00	70.20	
Li XX, 2015 [45]	Serum	Let-7c	Down		0.85 (0.79–0.91)	87.50	78.90	
Mangolini A (A), 2015 [46]	Serum	145	Down					
		425	Down					
		652	Down		0.83 (0.73–0.93)			
		10b	Up					
		148b	Down		0.74 (0.62–0.86)			
Mangolini A (B), 2015 [46]	Serum	145	Down					
		425	Down					
		652	Down		0.69 (0.58–0.80)			
		10b	Up					
		148b	Down		0.66 (0.51–0.80)			
Matamala N, 2015 [47]	Plasma	21	Up		0.61 (0.53–0.68)			
		96	Up		0.72 (0.65–0.78)	73.00	66.00	
		505	Up		0.72 (0.66–0.79)	75.00	60.00	
		125b	Up		0.64 (0.56–0.71)			
Shaker O, 2015 [48]	Serum	155	Up	39.57	0.99 (0.99–1.00)	94.10	100.00	39.57
		197	Up	29.80	0.98 (0.95–1.00)	95.30	100.00	29.80
		205	Up	27.48	0.99 (0.98–1.00)	98.80	100.00	27.48
		29b	Up	41.94	0.99 (0.98–1.00)	98.80	100.00	41.94
Zhang L, 2015 [49]	Serum	424	Up		0.75 (0.67–0.84)			1.77
		199a	Up		0.92 (0.87–0.96)			2.65
		29c	Up		0.72 (0.64–0.81)			1.97
Frères P, 2016 [50]	Plasma	16						1.70
		22						1.00
		103						0.80
		107						0.80
		148a						1.40
		19b						1.20
		Let-7d						0.90
		Let-7i						0.70
Fu L, 2016 [51]	Serum	184	Down	0.48	0.74 (0.66–0.82)	87.50	71.00	
		382	Up	1.32	0.90 (0.85–0.96)	93.00	75.00	
		598	Down	1.61	0.74 (0.66–0.82)	52.00	92.50	
		1246	Up	0.55	0.94 (0.90–0.98)	95.00	85.00	
Hamam R, 2016 [52]	Serum and Plasma	188	Up					
		1202	Up					
		1207	Up					
		1225	Up					
		1290	Up					
		3141	Up					
		4270	Up					
		4281	Up					
		642b	Up					
Hannafon BN, 2016 [53]	Plasma	21	Up		0.69 (0.49–0.89)			
		122	Up					
		1246	Up		0.69 (0.50–0.88)			
		Let-7a	Up					
Motawi TM, 2016 [54]	Serum	21		1.14	0.98 (0.96–1.00)	96.00	94.00	2.20
		221		1.21	0.97 (0.94–1.00)	92.00	88.00	2.09
Shimomura A, 2016 [55]	Serum	1246				88.30	93.40	
		1307				100.00	53.10	
		4634				3.40	73.60	
		6861				99.80	79.40	
		6875				14.70	76.80	
Thakur S, 2016 [56]	Serum	21	Up		0.79 (0.71–0.86)	88.00	65.00	
		145	Down		0.73 (0.66–0.81)	74.00	56.00	
		195	Down		0.80 (0.74–0.87)	77.00	71.00	
		210			0.64 (0.55–0.72)	78.00	61.00	
		221			0.63 (0.54–0.71)	65.00	57.00	
		Let-7a			0.76 (0.69–0.83)	71.00	67.00	
Gao S, 2017 [57]	Plasma	155	Up		0.77 (0.68–0.86)			
Zhang K, 2017 [58]	Blood	96	Up	2.73	0.77	53.00	100.00	
		182	Up	1.01	0.76	53.00	92.00	
		942	Up	1.04	0.81	67.00	100.00	
		30b	Up	2.04	0.93	80.00	100.00	
		374b	Up	1.52	0.82	87.00	69.00	
Heydari N, 2018 [59]	Serum	140	Up	0.13	0.67 (0.55–0.79)	70.00	50.00	
Zaleski M, 2018 [60]	Plasma	21			0.58 (0.46–0.71)			
		92			0.46 (0.33–0.60)			
		155			0.53 (0.36–0.69)			
		222			0.53 (0.40–0.67)			
		34a			0.72 (0.61–0.84)			
		Let-7c			0.51 (0.38–0.64)			
Kaharam M, 2019 [61]	Blood	101-3p						
		126-3p						
		126-5p						
		144-3p						
		144-5p						
		301a						
		664b						
McAnena P, 2019 [62]	Blood	195						0.73
		331						2.94
		181a						1.19
Peña-Cano MI, 2019 [63]	Serum	17	Up					0.50
		195	Up	0.04	0.88 (0.78–0.98)	83.30	78.30	4.33
		221	Down					0.70
Raheem AR, 2019 [64]	Serum	34a	Down	5.05	0.67 (0.53–0.81)	60.00	63.00	
Soleimanpour E, 2019 [65]	Plasma	21	Up		0.92			
		155	Up		0.99			
Anwar SL, 2020 [66]	Plasma	155	Up					
Arabkari V, 2020 [67]	Blood	16	Up		0.61 (0.47–0.76)			
		21	Up		0.65 (0.51–0.79)			1.35
		145	Up		0.83 (0.72–0.94)			1.61
		155	Up		0.76 (0.66–0.89)			1.63
		195	Down		0.81 (0.69–0.92)			0.14
		486	Up		0.90 (0.81–0.97)			2.25
		181a						1.52
		451a	Up		0.73 (0.61–0.86)			1.62
Ashirbekov Y, 2020 [68]	Plasma	16						0.69
		21						1.35
		29						0.98
		145						2.36
		191						1.87
		210						0.69
		222						0.98
Guo H, 2020 [69]	Plasma	21			0.66 (0.53–0.78)			
		1273g			0.63 (0.51–0.76)			
Holubekova V, 2020 [70]	Plasma	484	Up					1.10
		1260a	Up					1.22
		130a	Up					1.20
		99a	Up					1.33
Hosseini Mojahed FH, 2020 [71]	Serum	155	Up	1.40	0.89	77.80	88.89	1.00
Ibrahim AM, 2020 [72]	Plasma	21		4.94	0.78	63.30	100.00	
		145		0.78	0.70	45.00	83.30	
		10b		2.52	0.73	53.30	100.00	
		181a		1.51	0.70	50.00	80.00	
		Let-7		0.52	0.72	50.00	93.30	
Jang JY, 2020 [73]	Plasma	21	Down		0.92			
		24	Down		0.96	65.00	96.00	
		202	Down		0.86			
		206	Down		0.94	79.00	96.00	
		223	Down		0.81			
		373	Down		0.96			
		1246	Down		0.93	53.00	95.00	
		6875	Down		0.96	86.00	96.00	
		219b	Down		0.88			
Kim J, 2020 [74]	Plasma	202	Up	2.10	0.95 (0.88–1.02)	90.00	93.30	9.60
Pastor-Navarro B, 2020 [75]	Serum	21			0.77 (0.68–0.87)			
		155			0.32 (0.68–0.87)			
		205			0.65 (0.68–0.87)			
Bakr NM, 2021 [76]	Blood	373		360.00	0.98 (0.95–0.99)	90.80	98.40	
Diansyah MN, 2021 [77]	Plasma	21		1.66	0.92 (0.83–1)	92.30	81.20	4.36
Itani MM, 2021 [78]	Plasma	21	Up	4.46	0.76 (0.64–0.88)	73.00	81.00	
		139	Up	11.69	0.74 (0.62–0.87)	78.00	75.00	
		145	Up	10.18	0.78 (0.66–0.90)	83.00	78.00	
		155	Up	8.54	0.83 (0.71–0.95)	76.00	96.00	
		425	Up	9.09	0.81 (0.69–0.93)	78.00	91.00	
		451	Down	10.54	0.70 (0.57–0.83)	78.00	75.00	
		130a	Up	7.96	0.83 (0.72–0.94)	70.00	100.00	
		23a	Up	2.50	0.73 (0.61–0.85)	73.00	72.00	
Mohmmed EA, 2021 [79]	Serum and Plasma	106a	Up		0.95	83.00	100.00	3.63
Nashtahosseini Z, 2021 [80]	Serum	210	Up	0.82	0.72 (0.60–0.83)	68.00	51.00	2.72
		660	Up	0.77	0.77 (0.66–0.88)	79.00	61.00	2.71
Zhang K, 2021 [81]	Blood	185	Up	1.08	0.91 (0.83–0.99)	91.18	76.92	4.00
		362	Up	1.53	0.93 (0.88–0.99)	83.82	100.00	2.97
		106b	Up	1.26	0.82 (0.68–0.95)	79.41	76.92	1.89
		142-3p	Up	6.87	0.85 (0.76–0.98)	97.06	61.54	3.18
		142-5p	Up	1.60	0.85 (0.71–0.99)	85.29	76.92	2.46
		26b	Up	1.34	0.89 (0.81–0.97)	83.82	84.62	3.32
Zhao T, 2021 [82]	Serum	25	Up		0.75 (0.66–0.84)	57.10	95.00	
Li X, 2022 [83]	Serum	9	Up					
		17						
		148a	Up					
Liu H, 2022 [84]	Serum	103a	Up	3.40	0.70 (0.62–0.78)	78.20	74.70	
Mohamed AA, 2022 [85]	Serum	155	Up	7.50	0.94 (0.89–0.98)	86.90	90.00	
		373	Up	10.00	0.95 (0.90–0.98)	85.00	100.00	
		10b	Up	9.50	0.77 (0.69–0.84)	60.00	93.00	
		34a	Down	10.50	0.89 (0.82–0.94)	91.00	75.00	
Zavesky L, 2022 [86]	Plasma	451a	Down					1.39
		548b	Up					3.60
Zou R, 2022 [87]	Serum	24	Up		0.76			0.62
		324	Down		0.52			0.31
		377	Down		0.73			0.67
		497	Up		0.56			0.15
		125b	Up		0.58			0.13
		133a	Up		0.63			0.41
		19b	Down		0.63			0.26
		374c	Down		0.71			0.99

## Data Availability

Not applicable.

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
