# Peer review of "An Epidemiological Systematic Review with Meta-Analysis on Biomarker Role of Circulating MicroRNAs in Breast Cancer Incidence"

_ijms, 2023, doi:10.3390/ijms24043910_

Round 1
Reviewer 1 Report
The authors presented a meta-analysis of miRNA expression in breast cancer. The concept of the study is interesting, but the present study has some issues that need to be addressed.
In the "Materials and Methods" section the authors report on the collected data, mentioning that the miRNA expression levels were collected. First of all, why did they not collected clinical data, as they mention in the text for other factors such as obesity, smoking, working patterns etc. This would be interesting in order to find meta-analytic relations of these factors and as related to miRNA expression. The authors did not mention the method of miRNA detection, which is crucial regarding the expression of the molecules. It is completely different when comparing samples examined with micrarrays, NGS ot RT-PCR. Data are not comparable. In order to analyze such results the authors could have performed a data meta-analysis of either microarray or RT-PCR methodologies in order to be able to compare the use of miRNAs in breat cancer.
It would be more meaningful to compare gene expression in a cohort of works using bioinformatic methods than a systematic review approach.
Overall, the authors mention the use of miRNA on diagnosis, yet the question is confusing. The expression levels of miRNA probably are meant to be used as distinguishing biomarkers between tumor and controls, yet this question is not of much interest. It would be much more interesting to meta-analyze other parameters (clinical data) with respect to breast cancer.
It is understandable that the possible combinations of investigating miRNA expression with clinical data is large, and therefore I find the approach with not much interest. Thus, repeating my previous comment, the authors could have performed a bioinformatics analysis to analyze the amount of data collected.
Table 1 should be described were it is first referred. However, due ti its size the authors should consider moving it to the supplementary data.
The same applies for all figures. The text explaining each figure, should be close to the respective figure and not collect all figures at the end of the paper.
Author Response
We would like to thank you the reviewers for their valuable suggestions. We attached a new version of the paper with tracked changes of the text.
REV: In the "Materials and Methods" section the authors report on the collected data, mentioning that the miRNA expression levels were collected. First of all, why did they not collected clinical data, as they mention in the text for other factors such as obesity, smoking, working patterns etc.
ANSWER: Unfortunately, studies included in this review did not presented variables as obesity, smoking, SEP. The inclusion of these variables will be an important suggestion to improve the future research on this filed. We add a sentence in the discussion on this important issue.
REV: the authors did not mention the method of miRNA detection, which is crucial regarding the expression of the molecules. It is completely different when comparing samples examined with micrarrays, NGS ot RT-PCR. Data are not comparable. In order to analyze such results the authors could have performed a data meta-analysis of either microarray or RT-PCR methodologies in order to be able to compare the use of miRNAs in breast cancer.
ANSWER: to assure comparability of the data we included in this revi<ew only the validation phase of studies or studies on candidate miRNA (see page 4 line 142). As reported in discussion the method used to quantify circulating candidate microRNAs or to validate microRNAs is the quantitative reverse transcription (PCR RT-qPCR), only two studies have introduced the use of the digital.
REV: It would be more meaningful to compare gene expression in a cohort of works using bioinformatic methods than a systematic review approach.
ANSWER: bioinformatics methods would be useful to analyze primary data and in situation where the number of variables or the number of studies to manage were very high. In the case of this metanalysis we are handling secondary data, then we preferred to use the methods suggested by the Cochrane Handbook. Furthermore, unfortunately the number of studies for which it is possible to obtain the essential data to run the analyses and the variables reported for each subject are very scarce.
REV: Overall, the authors mention the use of miRNA on diagnosis, yet the question is confusing. The expression levels of miRNA probably are meant to be used as distinguishing biomarkers between tumor and controls, yet this question is not of much interest. It would be much more interesting to meta-analyze other parameters (clinical data) with respect to breast cancer.
ANSWER: This is another crucial point. All the studies published in literature are designed to measure the performance of miRNA as biomarkers to discriminate between cases and controls. We agree with the reviewer that this kind of study design is not useful to assess the possibility to use miRNA ad a diagnosis bopmarker neither to understand the role of miRNA on the carcinogenic pathway. We add a sentence in the discussion on this important issue.
REV: It is understandable that the possible combinations of investigating miRNA expression with clinical data is large, and therefore I find the approach with not much interest. Thus, repeating my previous comment, the authors could have performed a bioinformatics analysis to analyze the amount of data collected.
ANSWER: We included a new supplementary descriptive table on clinical data for each study. The message of the paper that we tried to clarify in this reviewed version, is that despite the large number of studies on the topic miRNA and breast cancer, the methodological problems of the studies, the scarcity of data reported in the articles and the great differences in materials and methods make the statistical synthesis of data not possible except for a few MicroRNAs in few studies. Thus, there is an urgent need for cohort-designed studies that follow standard and reproducible methods and data presentation rules.
REV: Table 1 should be described were it is first referred. However, due ti its size the authors should consider moving it to the supplementary data.
ANSWER: We need the permission of editorial staff to move table1 and figures, we will ask permission to move it in the proofreading phase if the article will be accepted.
Reviewer 2 Report
The manuscript entitled “An epidemiological systematic review with meta-analysis on biomarker role of circulating microRNAs in breast cancer incidence” by Padroni et al is an interesting manuscript that identifies circulating microRNAs related to breast cancer (BC) diagnosis post a systemic review and meta- analysis. Although it will be beneficial to predict biomarkers based on the review of existing literature, there are multiple complex factors that have to be carefully considered to derive at the results.
The authors have simplified complexities involving analysis of multiple studies in literature. However certain weaknesses as noted by authors are mentioned below-
1. Very few microRNAs were analyzed.
2. Inconsistencies were identified in circulating mircoRNAs in different studies.
3. Levels of microRNAs were different between sources such as serum, plasma and whole blood.
4. Data normalization can be a source of variability.
5. Appropriate sample size and control groups need to identified.
6. Genetic heterogeneity needs to be accounted.
More details of clinical support for the data need to be presented. In light of above weaknesses, authors should also explain in more detail why and how the chosen microRNAs were followed compared to others.
Author Response
We would like to thank you the reviewers for their valuable suggestions. We attached a new version of the paper with tracked changes of the text.
REV:
The authors have simplified complexities involving analysis of multiple studies in literature. However certain weaknesses as noted by authors are mentioned below-
- Very few microRNAs were analyzed.
- Inconsistencies were identified in circulating mircoRNAs in different studies.
- Levels of microRNAs were different between sources such as serum, plasma and whole blood.
- Data normalization can be a source of variability.
- Appropriate sample size and control groups need to identified.
- Genetic heterogeneity needs to be accounted.
ANSWER:
We simplify the complexities to try to harmonize very heterogenous studies in order to summarize the results.
- Very few microRNAs were analysed because they were the only miRNAs assessed in at least three different studies that reported the minimal core of data useful to perform a metanalysis.
- Inconsistencies were identified in circulating mircoRNAs in different studies: this is the message of this paper. There is an important inconsistence of microRNA among studies, that due the differences in methods and the weak study designs, it is not possible to understand if it is real.
REV:
More details of clinical support for the data need to be presented. In light of above weaknesses, authors should also explain in more detail why and how the chosen microRNAs were followed compared to others.
ANSWER:
We included a new supplementary descriptive table on clinical data for each study.
REV:
In light of above weaknesses, authors should also explain in more detail why and how the chosen microRNAs were followed compared to others
ANSWER:
Unfortunately, not all the studies on miRNA e breast cancer presented data on AUC and/or sensitivity and specificity. Meta-analysis was performed only for microRNAs analysed in at least three independent studies where sufficient data to make analysis were presented. The reason to include in the metanalysis only few microRNA is described at page 3 line 90 and page 4 line 171.
Round 2
Reviewer 1 Report
The authors have improved their manuscript. They have addressed my previous concerns. The authors should format their references according to the journal's guidelines.
The text in the boxes in figure 1, runs below the box-border.
Author Response
Thank you for the suggestions.
The reference format has been modified according to the journal's guidelines and figure 1 has been fixed.